# Pharmacokinetic/pharmacodynamic integration of amphenmulin: a novel pleuromutilin derivative against *Mycoplasma gallisepticum*

Wenxiang Wang,[1] Jiao Yu,[1] Xuan Ji,[1] Xirui Xia,[1] Huanzhong Ding[1]

**ABSTRACT**  Amphenmulin is a novel pleuromutilin derivative with great anti-mycoplasma potential. The present study evaluated the action characteristics of amphenmulin against *Mycoplasma gallisepticum* using pharmacokinetic/pharmacodynamic (PK/PD) modeling approaches. Following intravenous administration, amphenmulin exhibited an elimination half-life of 2.13 h and an apparent volume of distribution of 3.64 L/kg in healthy broiler chickens, demonstrating PK profiles of extensive distribution and rapid elimination. The minimum inhibitory concentration (MIC) of amphenmulin against *M. gallisepticum* was determined to be 0.0039 µg/mL using the broth microdilution method, and the analysis of the static time–kill curves through the sigmoid $E_{max}$ model showed a highly correlated relationship (R ≥ 0.9649) between the kill rate and drug concentrations (1–64 MIC). A one-compartment open model with first-order elimination was implemented to simulate the *in vivo* anti-mycoplasma effect of amphenmulin, and it was found that bactericidal levels were reached with continuous administration for 3 days at doses exceeding 0.8 µg/mL. Furthermore, the area under the concentration–time curve divided by MIC (AUC/MIC) correlated well with the anti-mycoplasma effect of amphenmulin within 24 h after each administration, with a target value of 904.05 h for predicting a reduction of *M. gallisepticum* by 1 $Log_{10}CFU/mL$. These investigations broadened the antibacterial spectrum of amphenmulin and revealed its characteristics of action against *M. gallisepticum*, providing a theoretical basis for further clinical development.

**IMPORTANCE**  *Mycoplasma* has long been recognized as a significant pathogen causing global livestock production losses and public health concerns, and the use of antimicrobial agents is currently one of the mainstream strategies for its prevention and control. Amphenmulin is a promising candidate pleuromutilin derivative that was designed, synthesized, and screened by our laboratory in previous studies. Moreover, this study further confirms the excellent antibacterial activity of amphenmulin against *Mycoplasma gallisepticum* and reveals its action characteristics and model targets on *M. gallisepticum* by establishing an *in vitro* pharmacokinetic/pharmacodynamic synchronization model. These findings can further broaden the pharmacological theoretical basis of amphenmulin and serve as data support for its clinical development, which is of great significance for the discovery of new antimicrobial drugs and the control of bacterial diseases in humans and animals.

**KEYWORDS**  *Mycoplasma gallisepticum*, pleuromutilin, pharmacokinetics, pharmacodynamics, *in vitro* dynamic model

M ycoplasma gallisepticum is currently an important pathogen in the global poultry industry that causes damage to the avian respiratory system and decreases meat and egg production (1), and has received long-term high attention from poultry farmers

Address correspondence to Huanzhong Ding, hzding@scau.edu.cn.

The authors declare no conflict of interest.

See the funding table on p. 13.

and researchers. Its infections frequently present predominantly as a chronic respiratory condition, but it also commonly gives rise to secondary infections by other pathogens like Newcastle disease virus, infectious bursal disease virus, and *Escherichia coli* that worsen the disease course (2). Moreover, there is considerable variability in the clinical symptoms resulting from different isolated strains (3), complicating epidemiological monitoring and control. The presently available commercial vaccines for *M. gallisepticum* principally comprise live attenuated forms as well as inactivated vaccines (4). However, owing to the complexity of strains and their capacity for immune escape, concerns remain over the potential for reversion to virulence and adverse effects with live vaccines, while inactivated vaccines are costly and involve intricate vaccination protocols (5, 6). Additionally, Mycoplasmas lack cell walls, conferring intrinsic resistance to cell wall synthesis inhibitors like β-lactams. Other mainstream effective drugs, such as quinolones and macrolides, have also shown a significant decrease in susceptibility to *M. gallisepticum* over long-term application (7). Therefore, the discovery and evaluation of new antimicrobial agents continue to be a sustained and crucial effort for researchers.

Pleuromutilin (Fig. 1 ) has attracted the investment of pharmaceutical workers due to its unique antimicrobial mechanism of action, targeting the peptidyl transferase center of the ribosome (8). Thus far, just four pleuromutilin derivatives have reached commercialization as final drugs—lefamulin (Fig. 1 and 2), retapamulin (Fig. 1 and 3), valnemulin (Fig. 1 and 4), and tiamulin (Fig. 1 and 5), the last two being restricted to veterinary applications. In recent years, numerous pleuromutilin derivatives have been designed globally using pleuromutilin as precursor structures (9). These derivatives are often reported to have excellent *in vitro* antimicrobial activities, but there have been limited publicly available results on their subsequent pharmaceutical development research and clinical evaluations. Amphenmulin (Fig. 1 and 6) was independently designed and synthesized in our laboratory. Previous studies have determined that it has better antibacterial activity against methicillin-resistant *Staphylococcus aureus* (MRSA) than tiamulin, and the acute toxicity tests have also proven it to be practically non-toxic (10, 11). Its pharmacological properties and application prospects deserve further exploration and development.

In the preclinical stage of innovative drugs, pharmacokinetic (PK) and pharmacodynamic (PD) studies are indispensable, serving as the cornerstone for subsequent clinical trials. PK/PD modeling integrates drug exposure levels and pharmacological responses, standing as a powerful tool for investigating dosing regimens and mitigating antimicrobial resistance risks (12). The European Medicines Agency (EMA) and the US Food and Drug Administration (FDA) have both released PK/PD modeling guidance, underscoring its importance in new drug research and development (R&D) (13, 14). In this study, we carried out pharmacokinetics of amphenmulin in broiler chickens, its pharmacodynamics against *M. gallisepticum*, and established *in vitro* dynamic modeling to obtain the PK/PD target indices, which further extended the antimicrobial spectrum of amphenmulin and enriched the theoretical basis of its action characteristics.

## MATERIALS AND METHODS

### Bacterial strain, reagents, and animals

The *M. gallisepticum* standard strain S6 was obtained from the National Center for Veterinary Culture Collection (Beijing, China). Amphenmulin (98%) was synthesized by our laboratory according to the published methods (10). Acetonitrile and all other liquid chemicals used were purchased from Macklin Biochemical Technology Co., Ltd (Shanghai, China). *M. gallisepticum* in this study was cultured under standard conditions of 37°C with the following nutrients: swine serum, *M. gallisepticum* artificial medium base from Qingdao Hope Biological Technology Co., Ltd (Qingdao, China), nicotinamide adenine dinucleotide (NADH) and cysteine from Aladdin Biochemical Technology Co., Ltd (Shanghai, China). Thirty healthy adult yellow-feathered broilers weighing 1.58 to 2.04 kg were used in this study, caged, fed without antibiotics and anticoccidial drugs, and

FIG 1 Structure of pleuromutilin (1), lefamulin (2), retapamulin (3), valnemulin (4), tiamulin (5), and amphenmulin (6).

watered and fed *ad libitum*. All animal experimental procedures were approved by the Animal Ethics Committee of South China Agricultural University (Approval number: 2023a015).

## *In vivo* pharmacokinetics

In this study, the dose design for the administration of amphenmulin was mainly based on the previous study report on tiamulin (15). The chickens were randomly divided into three groups (10 per group) and given a single dose of 20 mg/kg of body weight (bw.) of amphenmulin via intravenous, intramuscular injection, and oral gavage, respectively. Blood samples were collected from the brachial wing vein of each chicken at 0, 5, 10, 15, 30, 45 min, 1, 2, 3, 4, 6, 8, 10, 12, 24 h after intravenous injection, and at 0, 3, 6, 9, 12, 20, 39, 45 min, 1, 2, 4, 8, 12, 24 h after oral and intramuscular administrations. The blood samples were centrifuged at 3,500 rpm for 10 min. The plasma was collected and stored at −20°C until analysis.

After the plasma samples were restored to room temperature, 200 µL was taken and vortexed with 800 µL of acetonitrile, centrifuged at 12,000 rpm for 10 min. The supernatant was filtered through a 0.22-µm filter and analyzed using a high-performance liquid chromatography-tandem mass spectrometry (HPLC-MS/MS) system (Agilent 1200 HPLC and Applied Biosystem API4000 mass spectrometer, ABI Sciex, USA). Amphenmulin was separated on a C18 column (150 mm × 2 mm, 5 µm) at a flow rate of 0.5 mL/min, with a mobile phase of 0.1% aqueous formic acid and acetonitrile (40:60, vol/vol). The electrospray ionization (ESI) source of the mass spectrometer was operated in positive ion mode for multi-reaction monitoring, with the following ion source parameters: ion spray voltage 4,500 V, collision gas 5 psi, ion source gas1 50 psi, ion source gas2 55 psi, curtain gas 20 psi, source temperature 400°C. The precursor ion *m/z* of amphenmulin was 486.2, and the product ions *m/z* were 184.1 (quantifier) and 124.0, corresponding to collision energies of 8 and 48 V, respectively, with a declustering potential of 64 V, entrance potential of 10 V, and collision cell exit potential of 14 V. Importantly, the validation of the above method showed that the lower limit of quantification and limit of detection of amphenmulin in chicken plasma were 0.005 and 0.001 µg/mL, respectively, and the linearity was good in the concentration range of 0.005–0.5 µg/mL, with the correlation coefficients exceeding 0.99. Plasma spiked recoveries ranged from 86.00%

to 103.58%, with intra- and inter-day variations of 0.75%–8.46% and 3.15%–10.63%, respectively.

Additionally, the plasma concentration–time data of amphenmulin in each individual were further analyzed using the non-compartmental method in Phoenix 8.1 (Certara, USA) to calculate pharmacokinetic parameters, in which the area under the curve (AUC) was calculated using the log-linear trapezoidal method, and the results were reported as mean ± standard deviation (SD).

## Susceptibility assay

The minimum inhibitory concentration (MIC) of amphenmulin against *M. gallisepticum* S6 strain was determined with reference to the protocols previously reported (16, 17). The broth microdilution method was performed using 96-well plates to achieve a final titer of approximately $10^7$ colony-forming units (CFU)/mL of *M. gallisepticum* in each well. The final concentrations tested of amphenmulin ranged from $0.488 \times 10^{-3}$ to 0.125 µg/mL. Concurrently, a growth control (without drug), an end-point control (blank medium), and a sterile control (sterile broth) were also implemented. The minimum concentration that did not produce a color change in the wells after incubation was considered as the MIC value. For the agar dilution method, *M. gallisepticum* was inoculated and cultured on agar plates containing a twofold dilution series of amphenmulin concentrations, and the MIC was the lowest drug concentration that inhibited growth. Additionally, following the procedure of broth microdilution, the initial culture before incubation and the samples from wells where the color remained unchanged after 48 h of incubation were subjected to 10-fold serial dilutions. Each dilution sample (10 µL) was then plated on drug-free agar for colony counting. The lowest concentration that resulted in a bacterial reduction of greater than 3 $Log_{10}$CFU/mL was considered the minimum bactericidal concentration (MBC).

We also determined the minimum concentration inhibiting colony formation by 99% ($MIC_{99}$) of amphenmulin against the S6 strain by slightly modifying the existing methods (18, 19). The MIC measured by agar dilution was used as a baseline, and the concentration of the drug was linearly decreased by 10% to 0.5 MIC. By counting the colony growth of *M. gallisepticum* inoculated on these drug-containing agar plates and performing linear regression with drug concentration, we calculated the amphenmulin concentration that inhibits 99% growth of the S6 strain.

Furthermore, we determined the mutant prevention concentration (MPC) of amphenmulin against the S6 strain by agar dilution method (20). Briefly, *M. gallisepticum* at exponential phase was enriched to $\geq 10^9$ CFU/mL by centrifugation and cultured as described above. The lowest drug concentration that inhibited mycoplasma growth in the agar plate was determined as the primary MPC (MPCpr), and the determination was repeated after linearly reducing the MPCpr to 0.5 MPCpr at 10% drug concentration. The final mycoplasma-free growth concentration was the MPC. All the experiments of the above pharmacodynamic studies were conducted in triplicate.

## Static time–kill curves

Based on the MIC value from broth microdilution method, 7 mL blank medium, 0.2 mL amphenmulin solution, and 0.8 mL *M. gallisepticum* suspension were taken into 10-mL Celine bottles to make the final mycoplasma count of $10^7$ CFU/mL, as well as final concentrations of amphenmulin of 1, 2, 4, 6, 8, 16, 32, and 64 MIC, respectively. Growth control (without drug) and negative control (blank medium) were also set up for culture. Then at 0, 6, 9, 12, 24, 36, and 48 h, 100 µL of each culture was taken for counting using the aforementioned method, with a lower limit of 100 CFU/mL. The experiment was repeated three times.

The efficacy of amphenmulin against *M. gallisepticum* was measured by the kill rate per unit of time. Considering the growth of *M. gallisepticum* observed in preliminary experiments, this study selected seven time periods of 0–24, 0–36, 0–48, 6–24, 6–36, 6–48, and 12–48 h to calculate the kill rates of amphenmulin. The Sigmoid $E_{max}$ model in

Phoenix was used to fit the relationship between mean kill rate and drug concentration, which is shown below:

$$E = E_0 + \frac{E_{\max} \times C_e^N}{EC_{50}^N + C_e^N}$$

$E$, kill rate of amphenmulin against *M. gallisepticum* within a period; $E_0$, change rate of *M. gallisepticum* in the control group (without drug); $E_{\max}$, maximum kill rate within a period; $C_e$, concentration of amphenmulin; $EC_{50}$, drug concentration at half-maximal kill rate; $N$, Hill coefficient, reflecting the slope of the kill rate–concentration curve.

## *In vitro* dynamic model

According to previous reports (21–23), an *in vitro* model compatible with *M. gallisepticum* growth and amphenmulin pharmacokinetics was developed (24). It comprised a reservoir chamber containing fresh, drug-free medium; a reaction chamber with a three-necked flask where amphenmulin interacts with *M. gallisepticum*, containing 300 mL of medium in the outer chamber (external compartment, EC) and 10 mL of *M. gallisepticum* culture ($10^7$ CFU/mL) in a semipermeable membrane chamber (internal compartment, IC); and a waste chamber collecting effluent, which were connected via tubing and controlled by a peristaltic pump (Fig. 2). In this model, the flow rate was calculated as 1.63 mL/min from the reaction chamber volume and amphenmulin's elimination half-life ($T_{1/2Ke}$) after intravascular administration in chickens, enabling dynamic changes in drug concentrations that simulate amphenmulin elimination in chickens and its activity against *M. gallisepticum*. The *in vitro* elimination of amphenmulin followed first-order kinetics:

$$C = C_0 \times e^{-kt}$$

where $C_0$ is the initial amphenmulin concentration, $C$ is the amphenmulin concentration at time $t$, $k$ is the elimination rate constant, and $t$ is the time after dosing. Based on the maximum concentration reached in chickens after oral administration of amphenmulin, six dosing regimens of 0.1, 0.2, 0.5, 0.8, 1.0, and 1.5 µg/mL were designed, administered every 24 h for three times. By injecting the drug into both sides of the dialysis membrane simultaneously, the drug concentrations inside and outside the membrane reached equilibrium rapidly. Samples from the reaction chamber were collected at 0.083, 0.167, 0.25, 0.50, 0.75, 1, 2, 3, 4, 6, 9, 12, 24, 24.08, 25, 30, 36, 48.08, 54, 60, and 72 h after dosing for determination of amphenmulin concentration. Meanwhile, cultures from the IC were collected at 0, 3, 6, 9, 12, 24, 30, 36, 48, 54, 60, and 72 h after dosing for mycoplasma counting. Each sample was performed in triplicate.

The determination of amphenmulin in the medium referenced our laboratory's method for extracting tiamulin (25) and was also detected by the HPLC-MS/MS method mentioned in the PK section. The method validation results demonstrated that amphenmulin showed good linearity within the concentration range of 0.005–0.5 µg/mL ($R > 0.99$), blank medium spiked recoveries ranging from 86.54% to 100.55%, with intra- and inter-day variations of 1.58%–11.49% and 2.89%–10.49%, respectively, indicating the feasibility of the method.

## Integration and modeling of pharmacokinetics/pharmacodynamics

Phoenix software was used to analyze the *in vitro* concentration–time data of amphenmulin and calculate the AUC and peak concentration ($C_{\max}$), which were combined with MIC to obtain PK/PD parameters. Concurrently, the anti-mycoplasma effect was derived from the dynamic time–kill curve, which was analyzed with the PK/PD index using an inhibitory Sigmoid $E_{\max}$ model as follows:

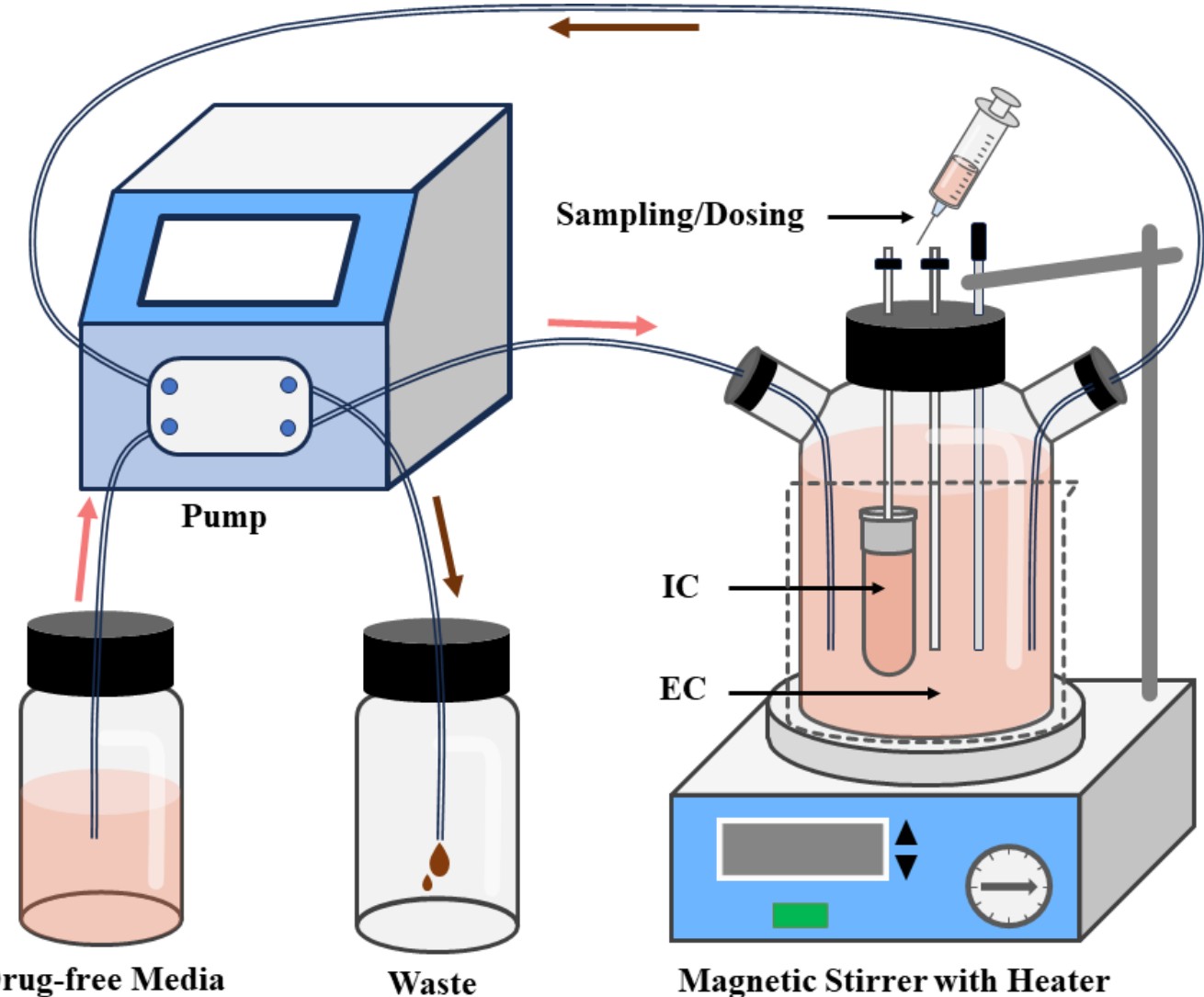

**FIG 2** The *in vitro* model simulating the effect of amphenmulin on *Mycoplasma gallisepticum* under the pharmacokinetic profile in chickens. IC, internal compartment; EC, external compartment.

$$E = E_0 - \frac{(E_0 - E_{\max}) \times C_e^N}{EC_{50}^N + C_e^N}$$

$E$, anti-mycoplasma effect, the change in *M. gallisepticum* counts during 24 h of cultivation; $E_0$, *M. gallisepticum* change in control (without drug); $E_{\max}$, maximal effect produced by amphenmulin against *M. gallisepticum* during 24 h; $C_e$, PK/PD parameters (AUC/MIC or $C_{\max}$/MIC); $EC_{50}$, the $C_e$ value needed to produce 50% of maximal effect; $N$, Hill coefficient, the slope of the fitted curve. Bacterial reduction ≥3 $Log_{10}$CFU/mL was defined as bactericidal effect; <3 $Log_{10}$CFU/mL was defined as bacteriostatic effect.

## RESULTS

### *In vivo* pharmacokinetics

The blood concentration–time curves of amphenmulin in chickens after intravenous, intramuscular, and oral administrations are shown in Fig. 3. The two extravascular routes reached peak levels before 30 min after administration, with $C_{\max}$ of 0.73 ± 0.36 and 1.93 ± 0.48 µg/mL for oral and intramuscular administrations, respectively. The main

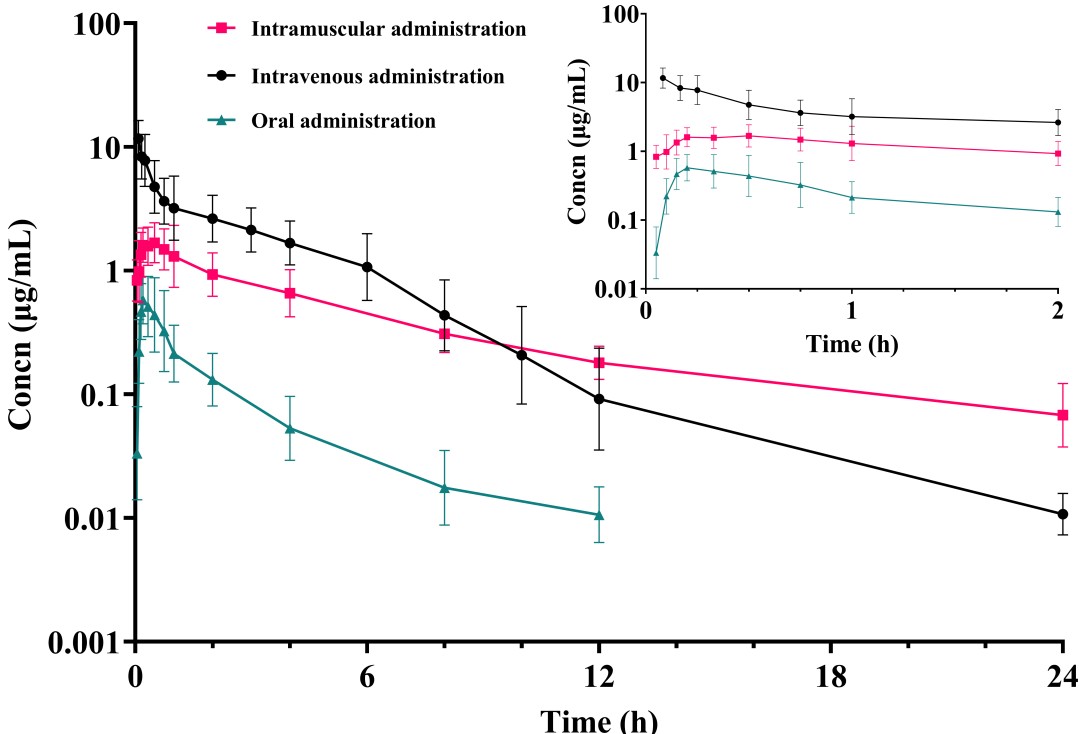

**FIG 3** Plasma concentration–time curves of amphenmulin (20 mg/kg of body weight.) in chickens after intravenous, intramuscular, and oral administrations. Data represent mean ± SD values for 10 chickens (logarithmic scale for the concentration; inset: visualization for 0–2 h).

**TABLE 1** Estimated pharmacokinetic parameters of amphenmulin [20 mg/kg of body weight (bw.)] in chickens after intravenous (IV), intramuscular (IM), and oral (PO) administrations[a]

| Parameter | IV | IM | PO |
|---|---|---|---|
| $C_{max}$ (µg/mL) | -- | 1.93 ± 0.48 | 0.73 ± 0.36 |
| $T_{max}$ (h) | -- | 0.38 ± 0.20 | 0.25 ± 0.11 |
| Kel (1/h) | 0.35 ± 0.11 | 0.13 ± 0.03 | 0.29 ± 0.08 |
| $T_{1/2Ke}$ (h) | 2.13 ± 0.63 | 5.50 ± 1.43 | 2.60 ± 0.87 |
| $AUC_{0-t}$ (h·µg/mL) | 18.76 ± 6.09 | 9.16 ± 2.65 | 1.06 ± 0.40 |
| $AUC_{0-\infty}$ (h·µg/mL) | 18.89 ± 6.03 | 9.85 ± 2.72 | 1.11 ± 0.43 |
| MRT (h) | 3.13 ± 1.05 | 7.89 ± 2.75 | 2.98 ± 1.05 |
| Cl (L/h/kg) | 1.17 ± 0.39 | -- | -- |
| $V_{ss}$ (L/kg) | 3.64 ± 1.58 | -- | -- |
| F (%) | | 52.14 | 5.88 |

[a]$C_{max}$, maximal concentration after administration; $T_{max}$, time to reach maximum concentration; Kel, elimination rate constant; $T_{1/2Ke}$, elimination half-life; $AUC_{0-t}$, area under the blood concentration–time curve from 0 to last measurable concentration; $AUC_{0-\infty}$, area under the blood concentration–time curve from 0 to infinity; MRT, mean residence time; Cl, body clearance; $V_{ss}$, apparent volume of distribution at steady-state; F, absolute bioavailability. Data represent mean ± SD values (n = 10).

pharmacokinetic parameters of amphenmulin in chickens are summarized in Table 1. From the results, the $T_{1/2Ke}$ of amphenmulin after intravenous and oral administrations was significantly lower than that of intramuscular injection. Based on the AUC of the three routes, the absolute bioavailability of oral and intramuscular injections was calculated to be 5.88% and 52.14%, respectively.

## Susceptibility assay

At an inoculum of $10^7$ CFU/mL, the MIC of amphenmulin for *M. gallisepticum* was determined to be 0.0039 and 0.0078 µg/mL using the microbroth dilution method

and agar dilution method, respectively. In addition, the $MIC_{99}$ and MBC were 0.0077 and 0.0156 µg/mL, respectively. For an inoculum of $10^9$ CFU/mL, the MPC value was 0.0500 mL, and a mutation selection window (MSW) of $0.77 \times 10^{-2}$–0.05 µg/mL can be derived for amphenmulin, which exhibits a degree of potential resistance risk.

## Analysis of static time–kill curves

The static time–kill curve of amphenmulin against *M. gallisepticum* strain S6 at 0–64 MIC drug concentrations are shown in Fig. 4a. Within 48 h of exposure of *M. gallisepticum* to amphenmulin, the mycoplasma counts only decreased by 0.44 $Log_{10}$CFU/mL at 1 MIC. At 2–32 MIC, the mycoplasma counts decreased by 0.57–2.7 $Log_{10}$CFU/mL, indicating a bacteriostatic effect of amphenmulin within this concentration range. However, when the concentration reached 64 MIC, the mycoplasma count decreased by 3.85 $Log_{10}$CFU/mL, exhibiting a bactericidal effect. These results indicated that the anti-mycoplasma effect of amphenmulin is enhanced with increasing static concentration *in vitro* (≤64 MIC).

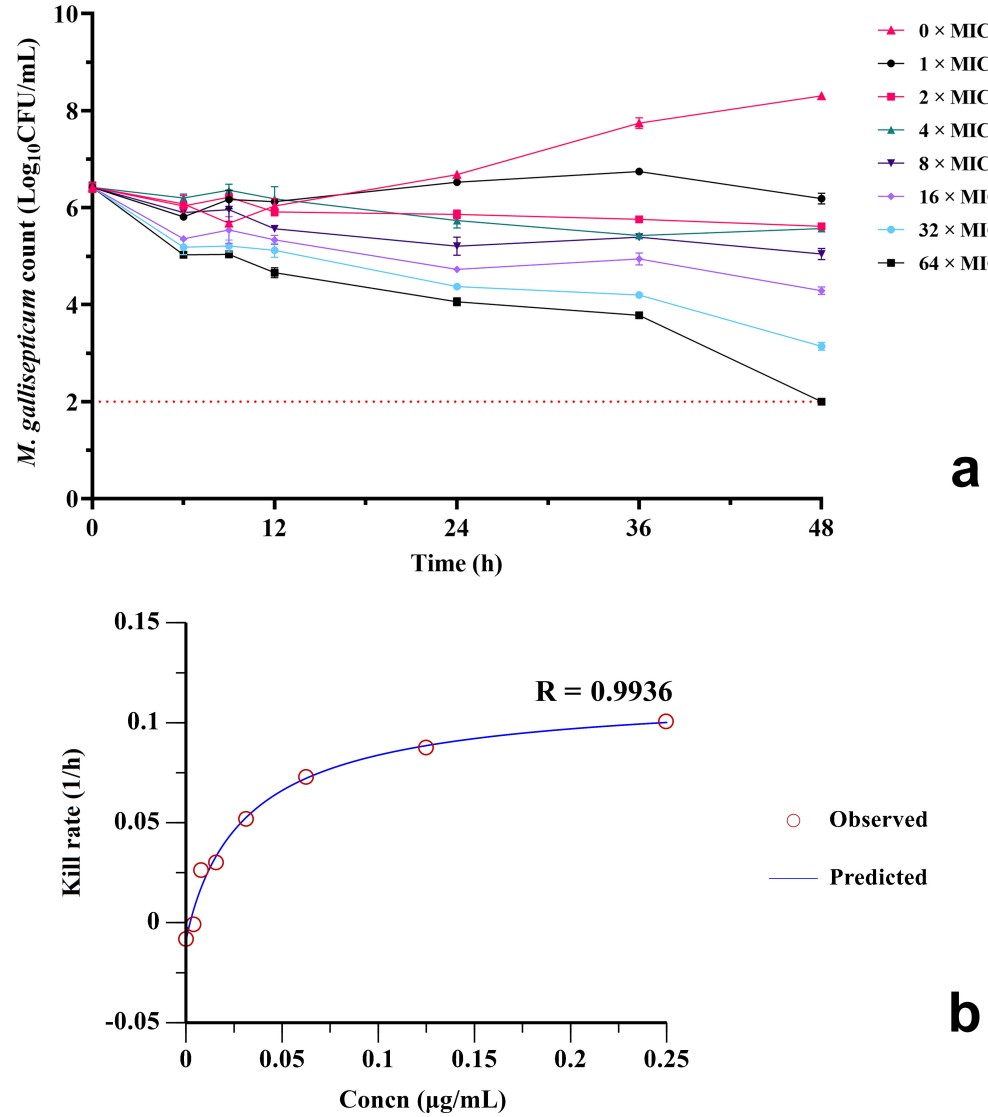

**FIG 4** (a) Static time–kill curve of amphenmulin against *Mycoplasma gallisepticum*. Minimum inhibitory concentration (MIC): 0.0039 µg/mL from broth microdilution method. (b) Best-fit curve from the $E_{max}$ model for *M. gallisepticum* exposure to different concentrations of amphenmulin over 0–24 h.

**TABLE 2** Estimated parameters of the kill rate of *M. gallisepticum* by amphenmulin from the $E_{max}$ model

| Time (h) | $E_{max}$ (1/h) | $EC_{50}$ (µg/mL) | $E_0$ (1/h) | Hill's slope | R |
|---|---|---|---|---|---|
| 0–24 | 0.1261 | 0.0325 | −0.0093 | 0.9238 | 0.9936 |
| 0–36 | 0.1817 | 0.0914 | −0.0359 | 0.4264 | 0.9856 |
| 0–48 | 0.3277 | 0.9135 | −0.0361 | 0.3760 | 0.9914 |
| 6–24 | 0.0805 | 0.0082 | −0.0373 | 2.7101 | 0.9748 |
| 6–36 | 0.0821 | 0.0053 | −0.0550 | 2.9068 | 0.9649 |
| 6–48 | 0.1743 | 0.0526 | −0.0524 | 0.4175 | 0.9820 |
| 12–48 | 0.2880 | 0.5245 | −0.0615 | 0.2919 | 0.9839 |

**TABLE 3** Estimated PK/PD parameters of amphenmulin against *M. gallisepticum* from the inhibitory $E_{max}$ model

| Parameter | $AUC_{24\,h}$/MIC | $C_{max}$/MIC |
|---|---|---|
| $E_{max}$ ($Log_{10}$CFU/mL) | −2.4214 | −3.3571 |
| $EC_{50}$ | 1,199.4720 | 441.4603 |
| $E_0$ ($Log_{10}$CFU/mL) | −0.3845 | −0.3575 |
| Hill's slope | 3.1997 | 1.3842 |
| R | 0.9657 | 0.8995 |

The mean kill rates and amphenmulin concentrations were fitted to the Sigmoid $E_{max}$ model to obtain parameters like $E_{max}$ and $EC_{50}$. The results in Table 2 indicate that the kill rates at different periods were well correlated with amphenmulin concentrations, with the goodness of fit (R) ranging from 0.9649 to 0.9936 and the maximum kill rate being 0.3277 1 /h. The model fitting within 0–24 h was optimal, and the relationship curve is visualized in Fig. 4b.

### *In vitro* pharmacokinetics and the effects on *M. gallisepticum*

In the one-compartment intravenous administration model simulated in this work, at doses of 0.1–1.5 µg/mL, the AUC at 0–24 h, 24–48 h, and 48–72 h were 0.34–6.19, 0.57–7.69, and 0.62–8.23 h·µg/mL, respectively, with measured $C_{max}$ after each administration of 0.12–1.52, 0.12–1.58, and 0.12–1.50 µg/mL, respectively. Throughout the experiment, the changes in drug concentration were higher than the MIC (0.0039 µg/mL) of amphenmulin against *M. gallisepticum* strain S6. The concentration-time curves of amphenmulin under different dosages are displayed in Fig. 5a.

The growth of *M. gallisepticum* exposed to dynamic drug concentrations can be seen in Fig. 5b. The decrease in *M. gallisepticum* counts within 24 h after each administration ranged from 0.15 to 2.08 $Log_{10}$CFU/mL, which was seen as a significant inhibition of proliferation. Within 0–72 h, the *M. gallisepticum* counts decreased by a maximum of 2.82 $Log_{10}$CFU/mL at doses of 0.1–0.5 µg/mL administered. When the dosage reached 0.8 µg/mL or above, amphenmulin exhibited bactericidal effects, and mycoplasma numbers at the highest dose fell below the counting limit (100 CFU/mL). The results demonstrated that the anti-mycoplasma effect of amphenmulin was positively correlated with the dosage.

### PK/PD modeling and analysis

Combining the *in vitro* PK data with the MIC value, the $AUC_{24\,h}$/MIC and $C_{max}$/MIC parameters of amphenmulin against *M. gallisepticum* can be obtained, and their relationship with the anti-mycoplasma effect is shown in Fig. 6, with $AUC_{24\,h}$/MIC demonstrating a higher correlation (R = 0.9657). The findings suggests that the effect of amphenmulin against *M. gallisepticum* is likely to be related to both the level and duration of drug exposure. The results of the inhibitory $E_{max}$ model can be seen in Table 3, which can be further calculated that the $AUC_{24\,h}$/MIC and $C_{max}$/MIC target values for a decrease of 1 $Log_{10}$CFU/mL of *M. gallisepticum* are 904.05 h and 190.11, respectively.

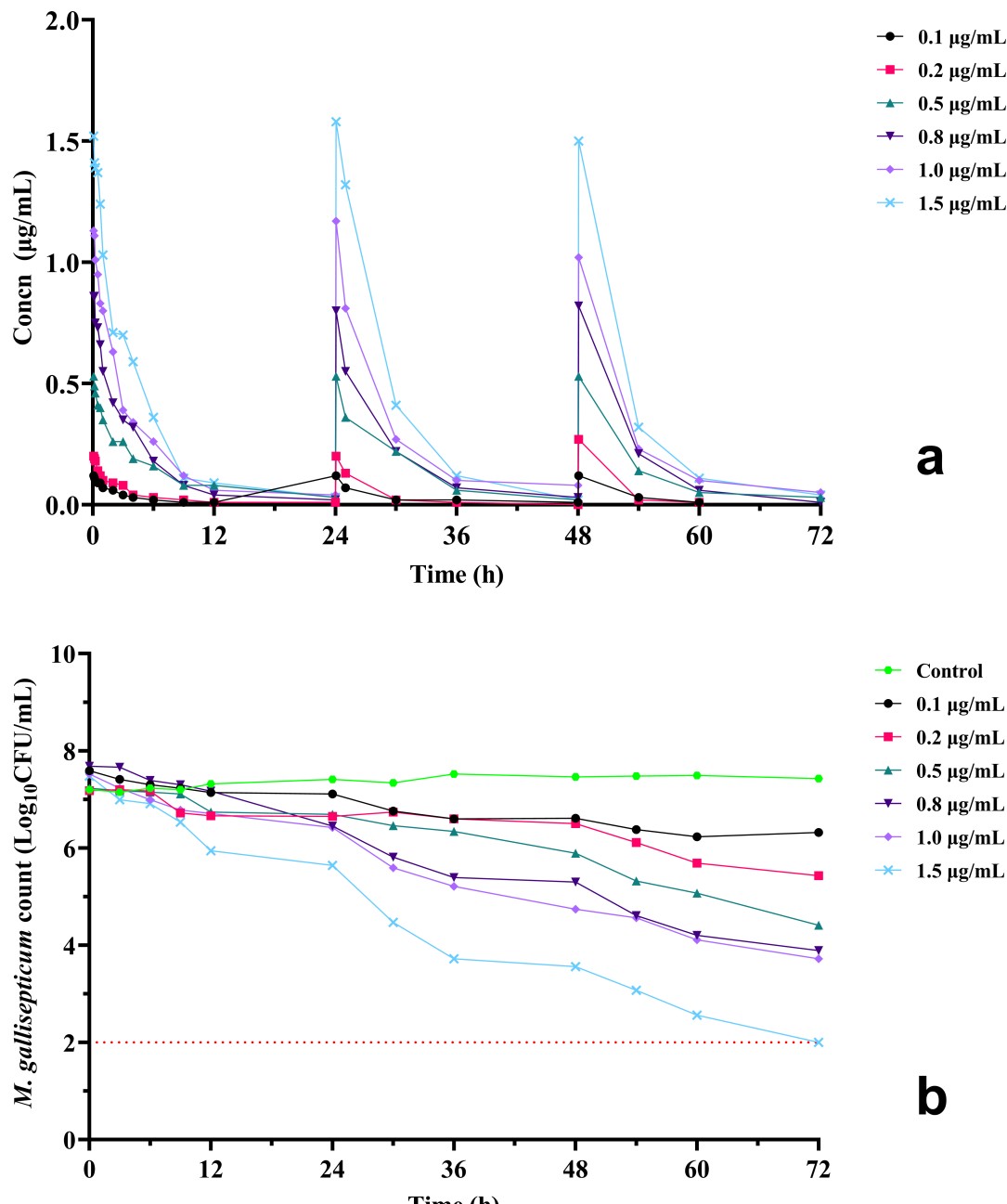

**FIG 5**  (a) Concentration–time curves of amphenmulin at different doses in the dynamic model. (b) Dynamic time–kill curves of various concentrations of amphenmulin against *Mycoplasma gallisepticum*.

## DISCUSSION

Currently, in veterinary medicine, antimicrobial drugs are confronted with the issues of increasing drug resistance, reduced efficacy caused by inappropriate use, and safety risks (26). Pleuromutilin mainly exerts antibacterial effects by inhibiting bacterial ribosomal protein synthesis and has low cross-resistance with other drugs; the resistance rates remain low after more than 30 years of use (27). In the development progress of this class of drugs, the structure–activity relationship determined that the optimization of pharmacological properties mainly focuses on the structural modification of the C14 side chain (28, 29), and amphenmulin was screened on this basis. A promising antimicrobial

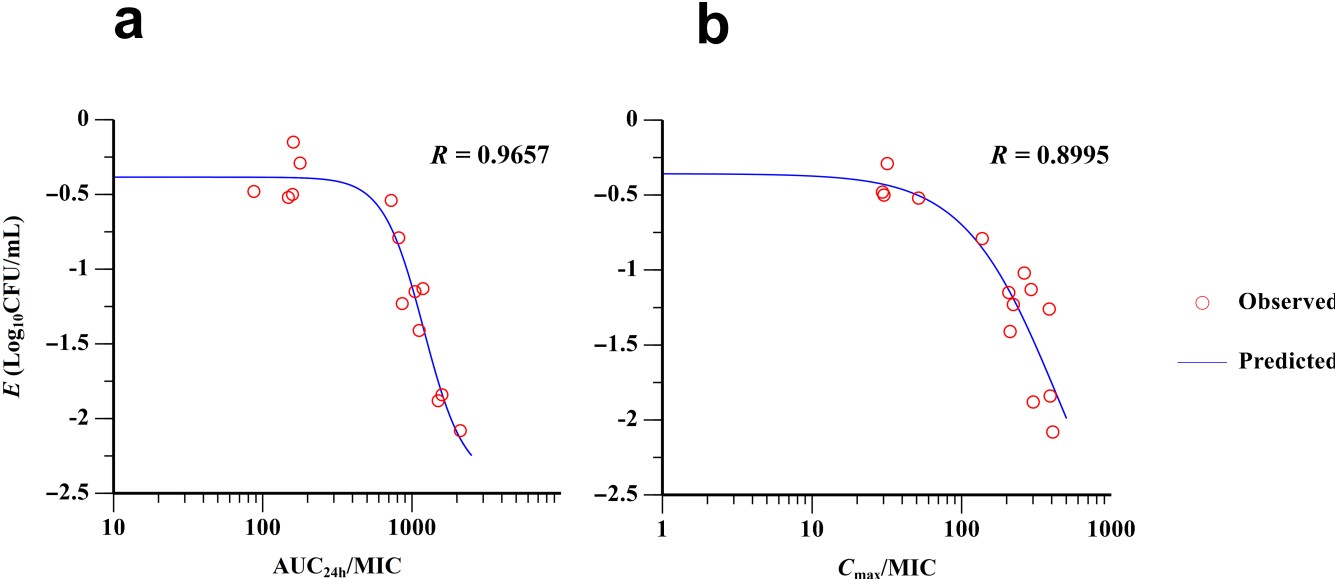

**FIG 6** Fitting of pharmacokinetic/pharmacodynamic parameters between anti-mycoplasma effects from the inhibitory $E_{max}$ model. (a):24 h area under the curve (AUC)$_{24\,h}$/minimum inhibitory concentration (MIC). (b): $C_{max}$/MIC.

agent should possess desirable pharmacokinetic and pharmacodynamic properties, which are the focus of preclinical research.

In our findings, rapid peak concentrations of amphenmulin were achieved in chickens after extravascular administration. However, in the intramuscular injection group, the $T_{1/2Ke}$ of amphenmulin was significantly longer than that in the intravenous and oral groups, exhibiting flip-flop kinetics, likely due to prolonged retention and slow release absorption from the injection site. As a typical pleuromutilin drug, high lipophilicity is a common trait (30, 31). This results in poor solubility of amphenmulin in the hydrophilic internal environment of organisms, making it difficult to efficiently penetrate biological membranes, leading to a low absorption rate. Similarly, the oral bioavailability of amphenmulin in chickens was only 5.88%, and 13.65% in mice (11). The complex gastrointestinal environment of animals means that only drugs with effective dissolution can be absorbed by the small intestine. Currently, the two pleuromutilin veterinary drugs are mainly administered as strong acid salts, such as tiamulin fumarate and valnemulin hydrochloride, which show bioavailability advantages among similar drugs (32–34). Additionally, the apparent volume of distribution ($V_{ss}$) of amphenmulin reached 3.64 L/kg, indicating extensive distribution *in vivo* (35), which coincides with the results obtained by Fu et al. in a tissue distribution study of another similar drug (36). Zeitlinger et al. also proved that after intravenous injection, the free drug concentration of lefamulin in the pulmonary epithelial lining fluid was 5.7 times higher than that in plasma, exhibiting good tissue penetration (37). Extensive tissue distribution has been validated among analogous agents, which is significant for systemic infectious disease treatment.

In pharmacodynamics, the MIC of amphenmulin against *M. gallisepticum* S6 strain was determined to be 0.0039 µg/mL by broth microdilution, half of the agar dilution result. This may be because agar dilution judges the MIC based on colony growth, while a small amount of *M. gallisepticum* may not noticeably change the broth color, causing misjudgment. However, this result is lower than the 0.03 µg/mL MIC of tiamulin against the same strain reported by Xiao et al. (38), indicating that S6 is more susceptible to amphenmulin. Additionally, we preliminarily evaluated the MSW of amphenmulin against *M. gallisepticum in vitro*. The MSW is generally between the MIC$_{99}$ and MPC. Bacteria exposed to drug concentrations within this window for extended periods have an increased risk of enriched resistant populations (39). The MPC of amphenmulin was

about 6.5 times the $MIC_{99}$ for *M. gallisepticum*, and fortunately, due to the sufficiently low value of MPC, the concentration of amphenmulin in chickens can easily exceed MPC for most of the time. Studies by Blondeau et al. showed that antibacterial effects at the MIC level are slow and incomplete (40). The MPC is a more valuable parameter than the MIC for preventing acquired resistance and optimizing dosing regimens (41, 42). Inadequately, the present study was conducted only with the standard strain of *M. gallisepticum*, and the susceptibility distribution of amphenmulin against different isolates remains to be further broadened and clarified.

Furthermore, static kill curves can more reliably demonstrate antibacterial effects compared to MIC measurement, as mathematical relationships are introduced to analyze PD parameters (43). Our results preliminarily proved that amphenmulin has significant inhibitory effects on *M. gallisepticum* growth, with a good correlation between kill rates and concentration in the 1–64 MIC range. However, this differs from the static kill curve of amphenmulin against MRSA, which shows more time-dependent tendency and bacteriostatic effects (11). This closely relates to the growth characteristics of different microorganisms and the action of the antimicrobial agents themselves. Under constant drug concentrations, the confrontation between drug efficacy and microbial regrowth capacity is intuitive, whereas in practical application, the ultimate effect is the combined interactions of the drug, pathogen, and host. Therefore, introducing dynamic PK/PD models is valuable, as it can more comprehensively describe the three-dimensional relationship between amphenmulin concentration, effect, and time against *M. gallisepticum* (44), and serve as a reference for *in vivo* studies, reducing costs of establishing animal models. Initially, we considered oral administration scenarios for amphenmulin, intending to build a one-compartment model with an additional absorption chamber to simulate the absorption process (45). However, we estimated the oral absorption rate constant ($K_a$) of amphenmulin to be approximately 3.15 1 /h based on the residual method, which would lead to a very small absorption compartment volume. Given the slow *in vitro* growth of *M. gallisepticum*, the rapid transient absorption process has little experimental impact, yet may cause errors. Therefore, building a one-compartment elimination model is more straightforward. The doses were referenced to the $C_{max}$ distribution after oral administration, and a multiple dosing strategy was implemented to offset the rapid elimination of amphenmulin, thereby allowing assessment of synchronous inhibitory effects on *M. gallisepticum* during the drug elimination phase.

For other similar drugs, Xiao et al. conducted PK/PD studies of tiamulin and valnemulin in an intratracheal infection model of *M. gallisepticum*, respectively (38, 46), and both demonstrated that the parameter most relevant to anti-mycoplasma effect was AUC/MIC, while optimized therapeutic dosages of 45 mg/kg and 6.5 mg/kg were given, respectively. It has also been reported that lefamulin showed time- and concentration-dependent activity against *Streptococcus pneumoniae* and *Staphylococcus aureus* in neutropenic murine thigh and lung infection model (47). Our study demonstrated that the $AUC_{24 h}$/MIC of amphenmulin showed significant correlation with anti-mycoplasma effect, and the target values required to reduce *M. gallisepticum* by 1 $Log_{10}$CFU/mL were estimated to be 904.05 h. Above the dose level of 0.8 µg/mL, amphenmulin could kill *M. gallisepticum*. These results reflect that the anti-mycoplasma effect of amphenmulin may be driven by both concentration and time, which could provide an important basis for optimizing the antimicrobial efficacy of amphenmulin and screening the appropriate dosage regimens. However, it remains to be determined whether amphenmulin is more time dependent against mycoplasma, as drug concentrations remained above the MIC throughout the multiple dosing in the dynamic model, which was also the main reason we did not examine the parameter $\%T > MIC$.

At present, the anti-mycoplasma activity of amphenmulin has been preliminarily demonstrated to be driven by the amount of drug exposure, which facilitates further assessment of its *in vivo* antimicrobial activity at a later stage. Although the observed pharmacokinetic profile of the raw amphenmulin in chickens is not ideal, as a highly permeable and poorly soluble compound, its *in vivo* kinetic processes can be optimized

through formulation design, crystal modification, optimal solvent selection, and other strategies. Alternatively, it can be further developed as a lipophilic precursor to enhance its bioavailability (48, 49).

## Conclusion

As a novel pleuromutilin derivative, amphenmulin exhibits rapid elimination kinetics and extensive tissue distribution in broiler chickens. *In vitro* studies show that *M. gallisepticum* is highly susceptible to amphenmulin, and both static and dynamic time–kill curves demonstrate a strong correlation between the effect of amphenmulin against *M. gallisepticum* and drug exposure. The $AUC_{24\ h}/MIC$ can serve as an important PK/PD target parameter to evaluate the anti-mycoplasma effect of amphenmulin, thus establishing a crucial foundation for subsequent drug efficacy studies.

## ACKNOWLEDGMENTS

The study was financially supported by the Guangdong Basic and Applied Basic Research Foundation (2022A1515010997).

H.D.: conceptualization, supervision, and project administration. J.Y. and X.J.: data curation and methodology. W.W. and X.X.: writing-original draft, review and editing. All authors contributed to manuscript revision, read, and approved the submitted version.

## AUTHOR AFFILIATION

[1]Guangdong Key Laboratory for Veterinary Drug Development and Safety Evaluation, College of Veterinary Medicine, South China Agricultural University, Guangzhou, China

## AUTHOR ORCIDs

Wenxiang Wang ⓘ http://orcid.org/0009-0002-9542-7746
Huanzhong Ding ⓘ http://orcid.org/0000-0002-0123-1228

## FUNDING

| Funder | Grant(s) | Author(s) |
|---|---|---|
| Guangdong Basic and Applied Basic Research Foundation | 2022A1515010997 | Huanzhong Ding |

## AUTHOR CONTRIBUTIONS

Wenxiang Wang, Writing – original draft, Writing – review and editing | Jiao Yu, Data curation, Methodology | Xuan Ji, Data curation, Methodology | Xirui Xia, Writing – original draft, Writing – review and editing | Huanzhong Ding, Conceptualization, Project administration, Supervision

## DATA AVAILABILITY

The data that support the findings of this study are openly available in figshare at http://doi.org/10.6084/m9.figshare.24721389

## ADDITIONAL FILES

The following material is available online.

Open Peer Review

**PEER REVIEW HISTORY (review-history.pdf).** An accounting of the reviewer comments and feedback.

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
