## [Reviewer comments · Microbiology Spectrum]

Microbiology Spectrum

Pharmacokinetic/pharmacodynamic integration of amphenmulin: a novel pleuromutilin derivative against *Mycoplasma gallisepticum*

Wenxiang Wang, Jiao Yu, Xuan Ji, Xi Xia, and Huanzhong Ding

Corresponding Author(s): Huanzhong Ding, South China Agricultural University

Review Timeline:

Submission Date:	October 14, 2023
Editorial Decision:	November 2, 2023
Revision Received:	November 16, 2023
Accepted:	November 20, 2023

Editor: Aude Ferran

Reviewer(s): Disclosure of reviewer identity is with reference to reviewer comments included in decision letter(s). The following individuals involved in review of your submission have agreed to reveal their identity: Andrew Mead (Reviewer #1)

Transaction Report:

DOI: <https://doi.org/10.1128/spectrum.03675-23>

Re: Spectrum03675-23 (Pharmacokinetic/pharmacodynamic integration of amphenmulin: a novel pleuromutilin derivative against *Mycoplasma gallisepticum*)

Dear Prof. Huanzhong Ding:

Thank you for the privilege of reviewing your work. Below you will find my comments, instructions from the Spectrum editorial office, and the reviewer comments.

Revision Guidelines

Sincerely,
Aude Ferran
Editor
Microbiology Spectrum

Reviewer #1 (Comments for the Author):

- Summary:
 - o The authors present a comprehensive PK/PD analysis of amphenmutilin, a novel pleuromutilin developed in their lab, against *M. gallisepticum*. The authors have done an excellent job covering the pharmacodynamics using both static and dynamic methods, albeit with the limitation of only a single strain. Furthermore, they present PK data for different dose administration

representing a well-rounded exploration of the PK in chickens. They have presented their findings clearly and discussed them with relation to other similar studies.

- Comments:

- o Introduction:

- Line 40: "... received long-term high attention from people."

- What people? The authors should specify where the interest is coming from, i.e. policy makers, poultry industry, health care sector, pharmaceutical industry, etc.

- o Methods:

- Line 87: The authors importantly ensured that the chickens in the Pk study were not fed with antibiotics. They should state if this also covers anti-coccidiostats.

- Line 116: Certara spelled incorrectly.

- Susceptibility testing was performed using an inoculum of 10^7 CFU/mL. The authors should justify why they have selected this inoculum size (i.e. to model disease populations), as there may be an inoculum effect with higher inoculum showing increased MIC. Note: CLSI and EUCAST perform MIC with a much lower standardised inoculum (10^5 CFU/mL).

- The authors have modelled both the PK, PD, and PK/PD using Phoenix software. It would be useful if they could provide the code used as a supplementary data file or provide an availability statement.

- o Results:

- Line 236: The range of Cmax was between 0.12 - 1.58 ug/mL it would be useful to remind the reader of the dosing profiles as this range is reporting the Cmax for 6 different doses. The authors could also comment on the accuracy of the in vitro model (i.e. dose 1.5 ug/mL gives Cmax = 1.58 ug/mL), is their variation between each of the three doses.

- Lines 252 - 23: The authors state that the effect of amphenmulin is concentration dependent. Although in the previous statement identify AUC/MIC (rather than Cmax) as the most relevant PK/PD index. This indicates both time and concentration are playing a role.

- Time is likely playing a significant role for this drug. Many drugs exhibiting protein synthesis inhibition show more time-dependent effect. This is also illustrated by the dynamic dosing response as compared to the static time-kill assays. In static curve over 24h bactericidal response is observed at 64x MIC. In dynamic curves to achieve a bactericidal response concentration over 0.8 ug/mL (100 - 200 x MIC) are required and this is over 72h, likely a result of the rapid elimination of drug reducing the time that the concentration is high above the MIC (i.e. reducing AUC).

- You also discuss the time and concentration dependency for tiamulin, valnemulin, and Lefamulin (see discussion). So, it is important to keep this consistent throughout.

- o Discussion:

- Line 261 - 263: Sentence structure is incorrect. Please revise.

- "The chemical structure modifications and bioactivity improvements focused on the C14 side chain, determined by structure-activity relationships of this class of drugs (28, 29), including amphenmulin."

- Line 291 - 293: Sentence should be simplified. Perhaps split into two sentences.

- Line 301: Typographical mistake: "Preliminarily"

- o Tables/Figures:

- Table 2: Indicates that susceptibility testing was performed at 10^7 CFU/mL. This is not the case for MPC which should be highlighted.

- Figure 2: Add the abbreviations for EC and IC for clarity.

- Figure 3: SD bars are only shown as positive (except for IV dose in 2h plot). Add full error bars on all graphs.

- Figure 4: Specify the MIC in the figure legend so its clear what concentrations are used for multiples of MIC. This is especially important to avoid confusion between MIC by broth dilution and agar dilution.

- Figure 5: As the bacterial curves in Fig. 5b relate to the dosing in Fig.5a I would recommend matching the colours of the curves to aid the reader.

Reviewer #2 (Comments for the Author):

The manuscript describes a new pleuromutilin derivative - amphenmulin. Its activity has been evaluated by the PK/PD approach. The study has additive value in the field of development of new antibacterial drugs for veterinary medicine. It contains new information. The experimental design is correct and gives the opportunity to obtain reliable results. The application of a dynamic model for evaluation of the activity of amphenmulin has additive value. The discussion is well written and properly reflects the obtained results. Despite the positive value of the manuscript, English language should be improved before acceptance.

Please, consider the minor remarks, listed below:

Abstract

It is very difficult to read the abstract due to the English language. The last sentence is not clear and it has to be revised.

Line 17: "druggability" is not very relevant term

Introduction

The introduction is well structured. It presents the problem in clear way.

Line 70: "PK/PD target parameters" should be replaced by "PK/PD target indices"

Materials and methods

Lines 90-91: "wing vein" - which vein? Give the name of the vein.

Line 98: Please, specify the model of HPLC-MS/MS.

Please, specify the value of LOD when the method of analysis is described.

Lines 111-113: Pharmacokinetic analysis should be better described. It was not explained if blood samples at every interval were obtained from every animal. Then, it is not clear if the PK parameters were calculated for every chicken. How AUC value was calculated, from 0 to 24 h? Which method was used for its calculation? What was the value of the extrapolation of AUC from t to infinity?

Line 119: Please correct "ug/mL"

Line 156: Is it necessary to write R (correlation coefficient between observed and model-predicted values) at the end of the equation. There is a coma before it and according to my knowledge, it is not a part of the equation.

Results

Figure 3 shows that concentration-time curves, which characterize the changes in the concentrations after i.v. and i.m. administration, are not parallel and flip-flop kinetics can be proposed.

Fig 6b shows that C_{max}/MIC is not a better surrogate marker for amphenmulin than AUC/MIC and it would be nice if authors mentioned this difference. Additionally, Fig. 6b shows that higher concentrations are necessary to analyze the relationship between the bacterial counts and C_{max}/MIC.

- **Summary:**

- The authors present a comprehensive PK/PD analysis of amphenmutilin, a novel pleuromutilin developed in their lab, against *M. gallisepticum*. The authors have done an excellent job covering the pharmacodynamics using both static and dynamic methods, albeit with the limitation of only a single strain. Furthermore, they present PK data for different dose administration representing a well-rounded exploration of the PK in chickens. They have presented their findings clearly and discussed them with relation to other similar studies.

- **Comments:**

- **Introduction:**

- Line 40: "... received long-term high attention from people."

What people? The authors should specify where the interest is coming from, i.e. policy makers, poultry industry, health care sector, pharmaceutical industry, etc.

- **Methods:**

- Line 87: The authors importantly ensured that the chickens in the Pk study were not fed with antibiotics. They should state if this also covers anti-coccidiostats.
- Line 116: Certara spelled incorrectly.
- Susceptibility testing was performed using an inoculum of 10^7 CFU/mL. The authors should justify why they have selected this inoculum size (i.e. to model disease populations), as there may be an inoculum effect with higher inoculum showing increased MIC. Note: CLSI and EUCAST perform MIC with a much lower standardised inoculum (10^5 CFU/mL).
- The authors have modelled both the PK, PD, and PK/PD using Phoenix software. It would be useful if they could provide the code used as a supplementary data file or provide an availability statement.

- **Results:**

- Line 236: The range of C_{max} was between 0.12 – 1.58 ug/mL it would be useful to remind the reader of the dosing profiles as this range is reporting the C_{max} for 6 different doses. The authors could also comment on the accuracy of the *in vitro* model (i.e. dose 1.5 ug/mL gives $C_{max} = 1.58$ ug/mL), is their variation between each of the three doses.
- Lines 252 – 23: The authors state that the effect of amphenmulin is concentration dependent. Although in the previous statement identify AUC/MIC (rather than C_{max}) as the most relevant PK/PD index. This indicates both time and concentration are playing a role.

Time is likely playing a significant role for this drug. Many drugs exhibiting protein synthesis inhibition show more time-dependent effect. This is also illustrated by the dynamic dosing response as compared to the static time-kill assays. In static curve over 24h bactericidal response is observed at 64x MIC. In dynamic curves to achieve a bactericidal response concentration over 0.8 ug/mL (100 – 200 x MIC) are required and this is over 72h, likely a result of the rapid elimination of drug reducing the time that the concentration is high above the MIC (i.e. reducing AUC).

You also discuss the time and concentration dependency for tiamulin, valnemulin, and Lefamulin (see discussion). So, it is important to keep this consistent throughout.

- **Discussion:**

- Line 261 – 263: Sentence structure is incorrect. Please revise.

“The chemical structure modifications and bioactivity improvements focused on the C14 side chain, determined by structure-activity relationships of this class of drugs (28, 29), including amphenmulin.”

- Line 291 – 293: Sentence should be simplified. Perhaps split into two sentences.
- Line 301: Typographical mistake: “Preliminarily”

○ **Tables/Figures:**

- Table 2: Indicates that susceptibility testing was performed at 10^7 CFU/mL. This is not the case for MPC which should be highlighted.
- Figure 2: Add the abbreviations for EC and IC for clarity.
- Figure 3: SD bars are only shown as positive (except for IV dose in 2h plot). Add full error bars on all graphs.
- Figure 4: Specify the MIC in the figure legend so its clear what concentrations are used for multiples of MIC. This is especially important to avoid confusion between MIC by broth dilution and agar dilution.
- Figure 5: As the bacterial curves in Fig. 5b relate to the dosing in Fig.5a I would recommend matching the colours of the curves to aid the reader.

Response to Reviewer 1 Comments

Dear Editors and Reviewer:

Thank you for your letter and for the reviewers' comments concerning our manuscript entitled "Pharmacokinetic/pharmacodynamic integration of amphenmulin: a novel pleuromutilin derivative against *Mycoplasma gallisepticum* (Spectrum03675-23)". Those comments are all valuable and very helpful for revising and improving our paper, as well as the important guiding significance to our researches. We have studied comments carefully and have made correction which we hope meet with approval. Revised portions are marked in red with track change mode in the "Marked-Up Manuscript". The main corrections in the paper and the response to the specific comments are as follows:

1. Line 40: "... received long-term high attention from people."

What people? The authors should specify where the interest is coming from, i.e. policy makers, poultry industry, health care sector, pharmaceutical industry, etc.

Response 1: We identified poultry practitioners and researchers as groups of concern for *M.gallisepticum* and added in Line 41.

2. Line 87: The authors importantly ensured that the chickens in the Pk study were not fed with antibiotics. They should state if this also covers anti-coccidiostats.

Response 2: We did not use any other antibiotics or anticoccidials during the experiment and feeding of the chickens, as stated in Lines 89-90.

3. Line 116: Certara spelled incorrectly.

Response 3: We apologize for the oversight in our writing, which has now been corrected.

4. Susceptibility testing was performed using an inoculum of 10^7 CFU/mL. The authors should justify why they have selected this inoculum size (i.e. to model disease populations), as there may be an inoculum effect with higher inoculum showing increased MIC. Note: CLSI and EUCAST perform MIC with a much lower standardised inoculum (10^5 CFU/mL).

Response 4: Regarding the inoculum in susceptibility testing, we had found no significant difference in the MIC of amphenmulin at inoculum levels of 10^7 , 10^6 , and 10^5 in early pre-experiments according to the guidelines published by Hannan^[1] and Tanner et al.^[2] Considering the slow growth of *M.gallisepticum* in vitro, a lower initial inoculum would have resulted in a prolonged MIC test period, and therefore we selected an inoculum of 10^7 for the subsequent experiments.

[1] Hannan, P. C. 2000. Guidelines and recommendations for antimicrobial minimum inhibitory concentration (MIC) testing against veterinary mycoplasma species. International Research Programme on Comparative Mycoplasmaology. Vet Res 31:373-95.

[2] Tanner, A. C., and C. C. Wu. 1992. Adaptation of the Sensititre broth microdilution technique to antimicrobial susceptibility testing of *Mycoplasma gallisepticum*. Avian Dis

36:714-7.

5. The authors have modelled both the PK, PD, and PK/PD using Phoenix software. It would be useful if they could provide the code used as a supplementary data file or provide an availability statement.

Response 5: Thanks to your suggestion, we have added an availability statement for the study data at the end of the manuscript.

6. Line 236: The range of C_{max} was between 0.12 - 1.58 ug/mL it would be useful to remind the reader of the dosing profiles as this range is reporting the C_{max} for 6 different doses. The authors could also comment on the accuracy of the in vitro model (i.e. dose 1.5 ug/mL gives C_{max} = 1.58 ug/mL), is their variation between each of the three doses.

Response 6: Following the reviewers' suggestions, we have reorganized the description of these results and emphasized the dose range of 0.1-1.5ug/mL (Line 246). Combined with the entire paragraph and with Figure 5a, we believe that the reader should be able to understand the drug concentration versus time profiles here.

7. Lines 252 - 23: The authors state that the effect of amphenmulin is concentration dependent. Although in the previous statement identify AUC/MIC (rather than C_{max}) as the most relevant PK/PD index. This indicates both time and concentration are playing a role. Time is likely playing a significant role for this drug. Many drugs exhibiting protein synthesis inhibition show more time-dependent effect. This is also illustrated by the dynamic dosing response as compared to the static time-kill assays. In static curve over 24h bactericidal response is observed at 64x MIC. In dynamic curves to achieve a bactericidal response concentration over 0.8 ug/mL (100 - 200 x MIC) are required and this is over 72h, likely a result of the rapid elimination of drug reducing the time that the concentration is high above the MIC (i.e. reducing AUC). You also discuss the time and concentration dependency for tiamulin, valnemulin, and Lefamulin (see discussion). So, it is important to keep this consistent throughout.

Response 7: We agree with the reviewer's comments about AUC/MIC, which does reflect the effect of the total drug exposure and is inextricably linked to both concentration and time. We have revised the inaccurate description (Line 265) and refined the discussion (Line 349).

8. Line 261 - 263: Sentence structure is incorrect. Please revise.

"The chemical structure modifications and bioactivity improvements focused on the C14 side chain, determined by structure-activity relationships of this class of drugs (28, 29), including amphenmulin."

Response 8: Thank you for the correction, we have revised the sentence (Lines 277-280).

9. Line 291 - 293: Sentence should be simplified. Perhaps split into two sentences.

Response 9: The original sentence structure has been simplified following the

reviewer's comment (see Line 312 of the revised manuscript).

10. Line 301: Typographical mistake: "Preliminarily"

Response 10: We are sorry for our oversight, the word has been corrected.

11. Table 2: Indicates that susceptibility testing was performed at 107 CFU/mL. This is not the case for MPC which should be highlighted.

Response 11: Thanks for pointing this out. And given that Table 2 only has one row of data, the authors have considered deleting Table 2 and using text descriptions (Lines 225-228) for the susceptibility results.

12. Figure 2: Add the abbreviations for EC and IC for clarity.

Response 12: We have added the IC and EC abbreviations to the legend of Figure 2 (Line 396).

13. Figure 3: SD bars are only shown as positive (except for IV dose in 2h plot). Add full error bars on all graphs.

Response 13: We have revised Figure 3 following the reviewer's comments.

14. Figure 4: Specify the MIC in the figure legend so its clear what concentrations are used for multiples of MIC. This is especially important to avoid confusion between MIC by broth dilution and agar dilution.

Response 14: Thank you for the correction, we have noted the specific MIC concentrations in the legend of Figure 4 (Lines 400-401).

15. Figure 5: As the bacterial curves in Fig. 5b relate to the dosing in Fig.5a I would recommend matching the colours of the curves to aid the reader.

Response 15: We have modified the color of the curves in Figure 5 following the reviewer's comments.

We appreciate for your warm work earnestly, and hope that the correction will meet with approval.

Once again, thank you very much for your comments and suggestions.

Response to Reviewer 2 Comments

Dear Editors and Reviewer:

Thank you for your letter and for the reviewers' comments concerning our manuscript entitled "Pharmacokinetic/pharmacodynamic integration of amphenmulin: a novel pleuromutilin derivative against *Mycoplasma gallisepticum*" (Spectrum03675-23). Those comments are all valuable and very helpful for revising and improving our paper, as well as the important guiding significance to our researches. We have carefully taken your comments into consideration and have made correction in preparing our latest revision, which we hope meet with approval. Revised portion are marked in red with track change mode in the "Marked-Up Manuscript". Below is our point-by-point response to the specific comments.

1. It is very difficult to read the abstract due to the English language. The last sentence is not clear and it has to be revised. Line 17: "druggability" is not very relevant term

Response 1: Following the reviewer's comments, we have optimized the writing of abstract and reorganized the description of the last sentence. In addition, we have also replaced and deleted the term "druggability" in the entire text.

2. The introduction is well structured. It presents the problem in clear way. Line 70: "PK/PD target parameters" should be replaced by "PK/PD target indices"

Response 2: Thank you for the correction, the word has been modified.

3. Lines 90-91: "wing vein" - which vein? Give the name of the vein.

Response 3: Blood samples were collected from the brachial wing vein, and we have added it in revised manuscript.

4. Line 98: Please, specify the model of HPLC-MS/MS.

Response 4: We have added specific instrument models in Lines 104-105.

5. Please, specify the value of LOD when the method of analysis is described.

Response 5: We have added the value of LOD (0.001 µg/mL) in Lines 114-115.

6. Lines 111-113: Pharmacokinetic analysis should be better described. It was not explained if blood samples at every interval were obtained from every animal. Then, it is not clear if the PK parameters were calculated for every chicken. How AUC value was calculated, from 0 to 24 h? Which method was used for its calculation? What was the value of the extrapolation of AUC from t to infinity?

Response 6: Thank you for your suggestion. Regarding the collection of plasma samples and the data analysis we did perform it on each chicken, for which we have improved the description in the revised manuscript (Line 97 and 119). In addition, we used the log-linear trapezoidal method built in Phoenix for AUC calculations. In Table 1, we give $AUC_{0-\infty}$ and AUC_{0-t} and their respective definitions, which are 2 important parameters specified by the FDA for the calculation of total exposure to drugs in vivo ([Bioavailability Studies Submitted in NDAs or INDe - General Considerations \(fda.gov\)](http://www.fda.gov/oc/ohrt/bioavailability_studies_submitted_in_ndas_or_inde.htm)).

<https://www.fda.gov/media/121311/download>) and the basis for our bioavailability calculations. The value of $AUC_{t-\infty}$ is not given because it was not used in the PK analysis, and the reader can also obtain it from the mathematical relationship between $AUC_{0-\infty}$ and AUC_{0-t} .

7. Line 119: Please correct "ug/mL"

Response 7: We are sorry for our oversight, the word has been corrected.

8. Line 156: Is it necessary to write R (correlation coefficient between observed and model-predicted values) at the end of the equation. There is a coma before it and according to my knowledge, it is not a part of the equation.

Response 8: We agree with this view and have removed R and its definition from the end of both equations in the manuscript.

9. Figure 3 shows that concentration-time curves, which characterize the changes in the concentrations after i.v. and i.m. administration, are not parallel and flip-flop kinetics can be proposed.

Response 9: We agree with this point. And we also emphasized that this may be related to the high lipophilicity of amphenmulin itself (Lines 285-289), especially for intramuscular injection, it is likely that the slow absorption of the drug has caused this "slow-release and long-lasting effect" phenomenon.

10. Fig 6b shows that C_{max}/MIC is not a better surrogate marker for amphenmulin than AUC/MIC and it would be nice if authors mentioned this difference. Additionally, Fig. 6b shows that higher concentrations are necessary to analyze the relationship between the bacterial counts and C_{max}/MIC .

Response 10: Thank you for your suggestion. The goodness of fit of AUC/MIC is relatively higher than that of C_{max}/MIC (based on the correlation coefficient R), which we mentioned in the Line 264 of the results section. The lack of higher concentrations is indeed influenced by the amount of data obtained from our research, as we designed the drug concentration in the dynamic model based on the peak concentration of amphotericin in vivo after oral administration. The high-dose group has already reached $500 \times MIC$, we believe that the test requirements have been met.

We appreciate for your warm work earnestly, and hope that the correction will meet with approval.

Once again, thank you very much for your comments and suggestions.

Re: Spectrum03675-23R1 (Pharmacokinetic/pharmacodynamic integration of amphenmulin: a novel pleuromutilin derivative against *Mycoplasma gallisepticum*)

Dear Prof. Huanzhong Ding:

Your manuscript has been accepted, and I am forwarding it to the ASM production staff for publication. Your paper will first be checked to make sure all elements meet the technical requirements. ASM staff will contact you if anything needs to be revised before copyediting and production can begin. Otherwise, you will be notified when your proofs are ready to be viewed.

Sincerely,
Aude Ferran
Editor
Microbiology Spectrum